# Laser Irradiation-Induced Pt-Based Bimetallic Alloy Nanostructures without Chemical Reducing Agents for Hydrogen Evolution Reaction

**Taiping Hu [1,2,†], Yisong Fan [3,4,5,†], Yixing Ye [1], Yunyu Cai [1], Jun Liu [1], Yao Ma [1], Pengfei Li [1,*] and Changhao Liang [1,2,*]**

[1] Key Laboratory of Materials Physics and Anhui Key Laboratory of Nanomaterials and Nanotechnology, Institute of Solid State Physics, Hefei Institutes of Physical Science, Chinese Academy of Sciences, Hefei 230031, China; htp821@mail.ustc.edu.cn (T.H.); yeyixing@issp.ac.cn (Y.Y.); jliu@issp.ac.cn (J.L.); yaoma0309@hotmail.com (Y.M.)

[2] Department of Materials Science and Engineering, University of Science and Technology of China, Hefei 230026, China

[3] Anhui Provincial Key Laboratory of Photonic Devices and Materials, Anhui Institute of Optics and Fine Mechanics, Hefei Institutes of Physical Science, Chinese Academy of Sciences, Hefei 230031, China; fan_ys01@163.com

[4] Advanced Laser Technology Laboratory of Anhui Province, National University of Defense Technology, Hefei 230037, China

[5] School of Chemistry and Materials Science, University of Science and Technology of China, Hefei 230026, China

\* Correspondence: pfli@issp.ac.cn (P.L.); chliang@issp.ac.cn (C.L.)

† These authors contributed equally to this work.

**Abstract:** Binary metallic alloy nanomaterials (NMs) have received significant attention because of their widespread application in photoelectrocatalysis, electronics, and engineering. Although various synthetic methods have been adopted to prepare binary alloy NMs, the formation of bimetallic alloy NMs by irradiating the mixed solutions of metal salts and metal powders, using a nanosecond pulsed laser in the absence of any reducing agent, is rarely reported. Herein, we report a simple method to fabricate PtX (X = Ag, Cu, Co, Ni) alloy NMs by laser irradiation. Taking PtAg alloys as an example, we present the growth dynamics of the PtAg alloys by laser irradiating a mixture solution of bulk Pt and AgNO₃. The experimental process and evidenced characterization indicate that the photothermal evaporation induced by laser irradiation can cause the fragmentation of the bulk Pt into smaller parts, which alloy with Ag atoms extracted from Ag⁺ by solvated electrons ($e^-_{aq}$) and free radicals ($H_{aq}$). These alloys were used as electrocatalysts for the hydrogen evolution reaction (HER), proving their potential application. Notably, in a 0.5 M $H_2SO_4$ solution, the PtNi alloy exhibited higher HER activity (44 mV at 10 mA/cm⁻²) compared to the untreated bulk Pt (72 mV). Our work provides unique insights into the growth processing of valuable Pt-based bimetallic alloy NMs by laser-assisted metallic alloying, which paves a path for the development of bimetallic alloy electrocatalysts.

**Keywords:** bimetallic alloy; laser irradiation; water decomposition; hydrogen evolution reaction

## 1. Introduction

The design of well-defined nanoscale bimetallic alloys shows great promise in modulating the physicochemical properties of materials for different applications [1–3]. They not only possess the physicochemical properties of each mono-elemental but also impart new physicochemical properties due to the synergistic interaction between the two monometallic counterparts [4–9]. For example, numerous reports have reported that the Pt-based bimetallic alloys exhibit outstanding catalytic performance, such as hydrogen evolution

reaction (HER) and oxygen reduction reaction (ORR), which is due to the ligand and strain effects caused by the introduction of different elements into the Pt matrix [10–13]. Therefore, many efforts have been devoted to preparing bimetallic alloy NMs, such as chemical co-reduction [14–16], electrochemical deposition [17], and microwave synthesis [18,19]. Although most of the conventional methods can prepare a variety of alloys, these manufacturing processes may require toxic reagents, expensive precursors, and surfactants, and involve complex ligand exchange reactions [20,21]. Additionally, the active surface of the synthesized NMs is often blocked because they have toxic reagents and residues on their surface, which prevents their use in biomedicine due to toxicity [22,23]. Therefore, the exploration of new synthesis technologies is desired to overcome these disadvantages.

Recently, laser in liquids synthesis of NMs has gained extensive attention due to its ability to create homogeneous, multi-component, nonequilibrium NMs with independently and precisely controlled properties, such as size, composition, morphology, and defect density [24]. In addition, laser in liquids synthesis is an environmentally friendly method since it avoids the use of additional surfactants and enables the rapid synthesis of NMs [4]. Therefore, numerous research groups have invested significant effort in the fabrication of NMs utilizing pulsed laser techniques, particularly for bimetallic nanoalloys [25–27]. For instance, Hu et al. prepared PtCo alloys by using laser ablation in a solution and investigated the formation mechanism [28]. Olea-Mejia, Oscar, et al. a adopted liquid laser ablation technique to obtain the AuAg alloys [29]. The aforementioned synthesis of bimetallic alloys is realized by means of a top-down method-laser ablation with a nanosecond laser. However, both Daria V. Mamonova and Maxim S. Panov utilized a laser-induced deposition approach to obtain bimetallic nanoparticles (NPs) [30,31]. Tibbetts et al. fabricated AuAg alloy NPs and investigated the formation mechanism through femtosecond laser co-reduction of $Au^+$ and $Ag^+$ ions in a solution. The reduction process was derived from the electrons originating from the femtosecond laser-induced decomposition of water [32]. Although the femtosecond laser-induced photochemical conversion of aqueous noble metal ions is well-established and widely used to fabricate noble nanoalloys, the corresponding nanosecond laser-based strategy is rarely reported. Exploring this issue is considered significant since it has the potential to provide us with good control over the formation process of NMs.

Herein, we present a novel strategy to synthesize PtX (X = Ag, Cu, Co, Ni) alloy NMs based on the nanosecond pulsed laser irradiation of an intermixture of Pt powder and relevant precursors without any reducing agent. The experimental process and evidenced characterization indicate that laser-induced water decomposition produces solvated electrons ($e^-_{aq}$) and free radicals ($H_{aq}$) that act as reducing agents for Ag atoms nucleated from $Ag^+$. Simultaneously, the bulk Pt is broken into smaller parts which keep zero chemical states and alloy with Ag atoms. The obtained bimetallic nanoalloys were used as electrocatalysts for the HER in acidic media. In particular, the PtNi alloy showed higher HER activity (44 mV at 10 mA/cm$^{-2}$) in a 0.5 M $H_2SO_4$ solution compared to the untreated bulk Pt (72 mV). This work provides a viable way to explore bimetallic alloy electrocatalysts for HER.

## 2. Results and Discussion

### 2.1. Structural and Morphological Characterizations

A facile procedure was proposed to prepare the PtAg alloys (the details are shown in the experimental section). As shown in Figure 1a, the bulk Pt and $AgNO_3$ were mixed to form a homogenous solution, which was subsequently irradiated by the pulsed laser. To investigate the crystal structure of the formed PtAg alloys, powder X-ray diffraction (XRD) was employed for characterization. The XRD patterns of Pt, as depicted in Figure 1b, were consistent with the standard diffraction of JCPDS:04-0482. Specifically, the cubic phase of Pt exhibited diffraction peaks at 39.89°, 46.3°, and 67.9°, which corresponded to (111), (200), and (220) crystal planes. Based on Vegard's law, the diffraction peak shift in the crystalline structure of an alloy either increases or decreases, depending on the concentration ratio of its constituent elements [33]. Figure 1b indicates that the XRD diffraction peaks of

the prepared PtAg alloys negatively shifted to an angle with respect to pure Pt, which revealed the success of alloying. The low-magnification transmission electron microscopy (TEM) image of the PtAg alloys in Figure 1c showed a chain width of approximately 10 nm. As depicted in Figure 1d, the lattice fringe of 0.231 nm corresponding to the (111) plane was observed in the high-resolution TEM (HRTEM) image of the PtAg alloys. The HRTEM image in Figure 1d also revealed the presence of twin boundaries in the PtAg alloys due to the non-equilibrium condition in the liquid environment, which may provide an opportunity for designing other defect-rich alloy materials. The atomic structure exhibited symmetric lattice arrangements on both sides of the coherent twin boundary, which was clearly observed in the HRTEM image. The corresponding EDX energy spectrum, EDX line scans, and EDS mapping images suggest that the Pt and Ag atoms had an atomic ratio of 57.4:42.6 and were uniformly distributed throughout the PtAg alloys (Figure 1e–g). The aforementioned results exhibited the successful formation of PtAg alloys.

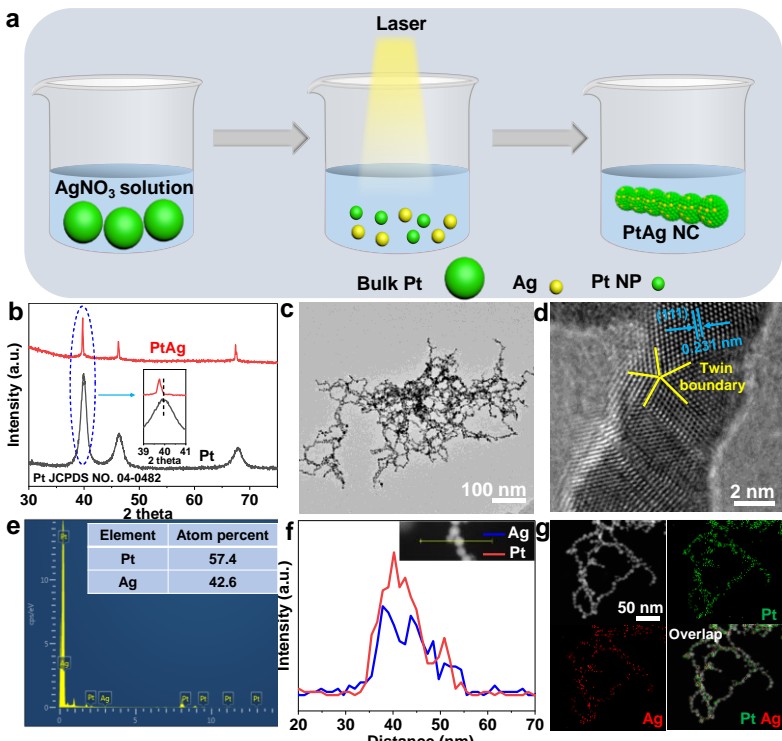

**Figure 1.** (**a**) Schematic diagram of the synthesis of the PtAg alloys, (**b**) XRD patterns of Pt and the PtAg alloys, (**c**) TEM image of the PtAg alloys, (**d**) the HRTEM image of the PtAg alloys, (**e**) EDX energy spectrum of the PtAg alloys, and (**f,g**) EDX line scans and EDS mapping of the PtAg alloys.

To gain further insights into the oxidation state and surface composition of the synthesized PtAg alloys, an XPS study was conducted on the PtAg samples, and the corresponding spectra are depicted in Figure 2. The XPS spectra of all samples confirmed the presence of Ag and Pt elements in the PtAg alloys and Pt, respectively (Figures S1a,b). The C1s spectra of all samples, as depicted in Figure S1c, exhibited a major peak at 284.8 eV with the same intensity, which confirmed a homogeneous amount of carbon in all the samples [34]. Through XPS analysis in Figure 2a, the observed peaks at 71.1 and 74.4 eV corresponded to metallic $Pt^0$ in the PtAg alloys but showed a negative shift compared to Pt, suggesting a charge transfer between Ag and Pt via electron interactions, which corresponded to their Pauling electronegativity scale [35]. In Figure 2b, the peaks at 367.7 and 373.8 eV were ascribed to metallic $Ag^0$ [36]. In addition, in Figure S1d, the auger electron spectrum of Ag for the PtAg alloys further confirmed the existence of metallic Ag in the PtAg alloys [37]. Based on these results, it can be confirmed that the PtAg alloys consist mostly of metallic Pt and Ag.

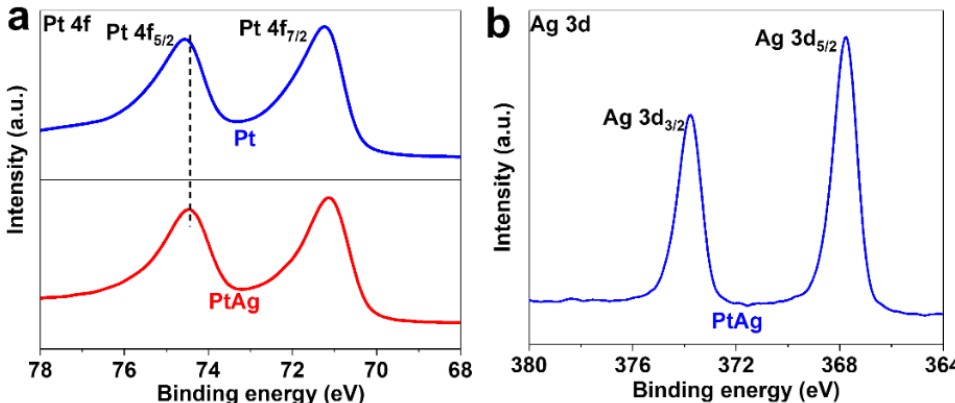

**Figure 2.** (**a**) XPS spectra of Pt 4f for the PtAg alloys and Pt, (**b**) XPS spectra of Ag 3d for the PtAg alloys.

One pressing question that needs attention is the formation process of the PtAg alloys, which was investigated by analyzing the TEM images of the samples at different laser irradiation times (10 s, 1 min, and 5 min). Figure 3 displayed the representative TEM image and line scan at various locations of the PtAg alloys that were synthesized using the laser irradiation procedure under different irradiation times. After irradiation for 10 s, small-sized PtAg alloys appeared. However, the Pt and Ag elements were not evenly distributed in different positions. It is crucial to note that the morphology of the products underwent a significant change when compared to the unirradiated initial bulk Pt. This suggested that the laser-induced alloying of the bulk Pt and $Ag^+$ was exceptionally fast, taking less than 10 s. As the irradiation time increased, the morphology of the PtAg alloys remained relatively unchanged as the irradiation time was less than 1 min. However, the distribution of Pt and Ag elements at various locations was nearly identical, indicating that laser irradiation caused the elements to be redistributed. After irradiation for 5 min, the distribution of Pt and Ag elements at various locations was nearly identical and almost the same as that in the sample under irradiation for 1 min (Figure 3b,c). The aforementioned results indicated that during the process of laser-induced reduction of $Ag^+$ ions to form the PtAg alloys, the initial stage was accompanied by changes in size and morphology, as well as the reduction of $Ag^+$ ions. In the subsequent stage, laser irradiation induced the redistribution of elements and reached a steady state.

The optical absorption spectra of the mixture of the bulk Pt and $AgNO_3$ solution before and after laser irradiation at different times were exhibited in Figure 4. Before the laser irradiation, the adsorption spectrum of the mixed solution almost had no SPR peak because the bulk Pt is the main content. Generally, the plasmon peak profile of noble metals, such as gold and silver, is highly dependent on the size and shape of the metal NPs. This is owing to the plasmon resonance, which is responsible for the unique optical properties of these materials and strongly influenced by the collective oscillation of electrons within the metal particles [38]. In comparison to the Pt colloids, the mixed solution demonstrated a wider plasmon peak and an increase in absorption density range from 250 to 400 nm after 10 s of irradiation and gradually showed a significant increase in absorption density with increasing the irradiation time. However, it is important to consider the hydrodynamic diameter of colloids in solution, as they tend to aggregate to some extent due to the dominance of surface double-layer structures [26]. According to Figure 1c, it was found that the PtAg alloy possessed the nanochains structure, which may account for the observed increase in optical absorption intensity within the 250–400 nm range. As the irradiation time increased, the corresponding optical images showed a gradual darkening of the sample color, which also confirmed the occurrence of the alloying process.

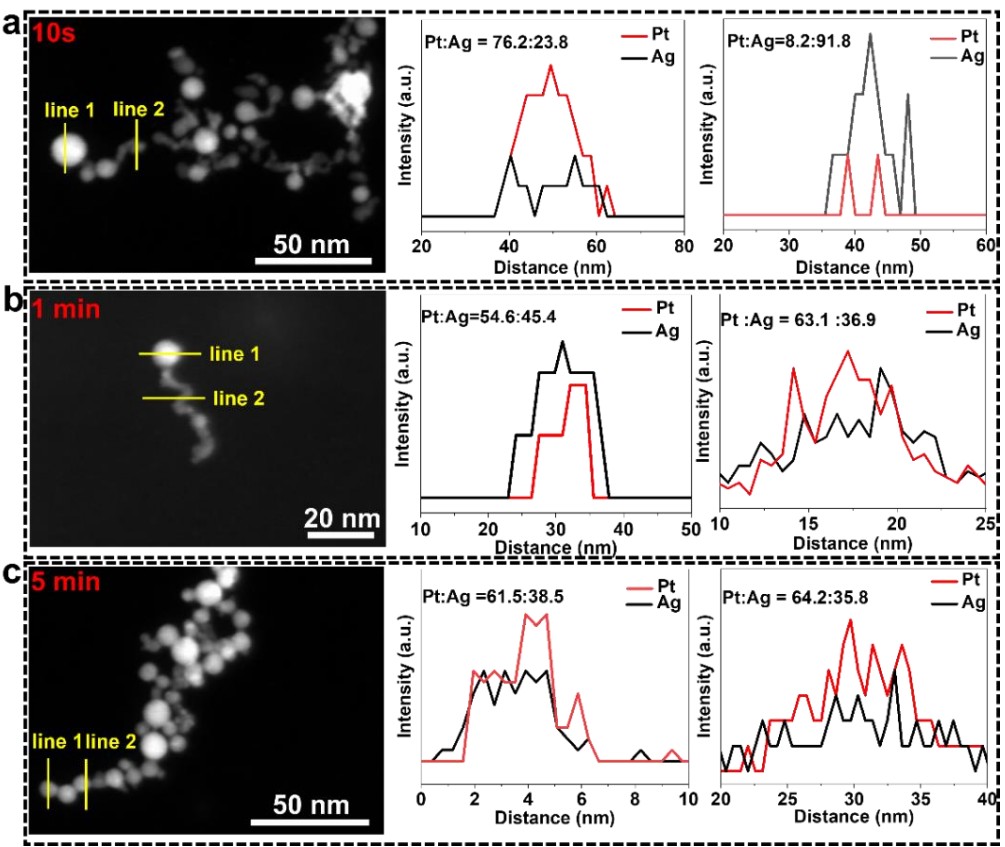

**Figure 3.** TEM images and line scan images of the bulk Pt and AgNO$_3$ solution: (**a**) 10 s, (**b**) 1 min, and (**c**) 5 min after laser irradiation.

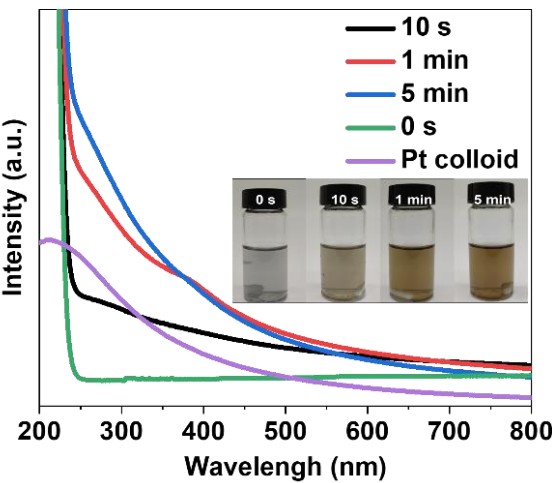

**Figure 4.** UV-vis absorption spectra and the corresponding optical image (inset) obtained from the mixture of bulk Pt and AgNO$_3$ solution and the bulk Pt.

## 2.2. Formation Mechanism of PtAg Alloys

Finally, the formation mechanism of the PtAg alloys was proposed. As shown in Figure 5a, firstly, when the bulk Pt absorbed pulsed laser energy in a liquid environment, it underwent a phase transition. At the same time, the temperature of the bulk Pt exceeded the boiling point of the surrounding liquid, thereby leading to the evaporation of the liquid and the formation of a shell around the bulk Pt with high vapor pressure. This asymmetry in vapor pressure may cause the internal molten NPs to break into smaller parts (Figure 5b,c) [24,39,40]. Additionally, at the same time, the energy density under

the laser beam was high enough to induce the decomposition of water molecules, thus yielding solvated electrons ($e^-_{aq}$) and free radicals ($OH_{aq}$ and $H_{aq}$) [41–43]. Hence, the $e^-_{aq}$, and $H_{aq}$ radicals acted as potential reducing agents for the reduction of $Ag^+$ ions with redox potentials of $E^0$ ($H_2O/e^-_{aq}$) = −2.8 V vs. SHE and $E^0$ ($H^+/H$) = −2.3 V vs. SHE, respectively. Despite a very short lifetime, both species were capable of reducing the $Ag^+$ ions to zero valences by the following equation:

$$H_2O \rightarrow e^-_{aq} + H_{aq} + OH_{aq} + \dots \tag{1}$$

$$Ag^+ + e^-_{aq} \rightarrow Ag^0 \tag{2}$$

$$Ag^+ + H_{aq} \rightarrow Ag^0 + H^+ \tag{3}$$

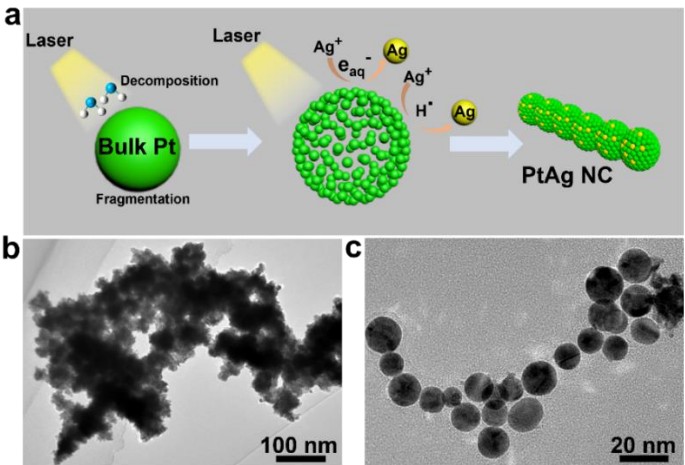

**Figure 5.** (**a**) Schematics of the alloying process of PtAg alloys, (**b**) the bulk Pt, (**c**) Pt NPs (obtained through a fragment of the bulk Pt).

Thus, the smaller Pt species alloyed with the Ag atoms during the laser irradiation. To further understand the formation mechanism of the PtAg alloys, a series of control experiments were conducted. As shown in Figure S2, the Ag NPs can also be obtained through laser irradiation of the $AgNO_3$ solution without the addition of a bulk Pt, which was consistent with the UV-visible spectra shown in Figure S3. In contrast, when a 300 W Xenon lamp with a wavelength of 520 nm was used to irradiate the $AgNO_3$ solution, there was no change in the UV-visible spectra (Figure S4), indicating that the laser played a crucial role in decomposing water to generate reducing species. Furthermore, both the $AgNO_3$ solution and the mixture of the bulk Pt with the $AgNO_3$ solution showed a significant drop in pH (Figure S5), indicating the generation of $H^+$ ions in the solution. Additionally, to demonstrate the alloying process of the Ag atoms with the Pt atoms, the mixture of the bulk Pt and Ag colloids was irradiated under the same condition. A 15 nm-sized Ag NPs was generated via laser ablation of the Ag plates in deionized water (Figure S6). Following this, the PtAg alloy NPs were produced by irradiating a mixture of 2 mg bulk Pt and 2 mL of Ag colloid. As shown in Figure S7, TEM-EDX line scans and EDS mapping images indicated that the spherical components consisted of PtAg alloys, suggesting that the laser played a role in the alloying process after the $Ag^+$ ions were reduced to Ag.

To test the universality of this synthesis method for preparing the Pt-based bimetallic alloys, the PtX (X = Cu, Co, Ni) alloys were successfully fabricated using the same procedure. Details of the morphological and compositional information of these alloys were shown in Figures 6 and S8. As shown in Figure 6a, after the irradiation of a mixed solution containing the bulk Pt and $Cu(NO_3)_2$, spherical NPs were obtained. The corresponding EDS mapping images showed the Pt and Cu elements were evenly distributed throughout the PtCu NPs.

In addition, The EDS line scans and EDX spectra produced by the prepared spherical NPs (Figure S8a,b) indicated the formation of uniform two-component particles, which consisted of Pt and Cu elements with an atomic ratio of 0.94:0.06. The above characterization indicated the successful formation of the PtCu alloy. By changing the $Cu(NO_3)_2$ to $Co(NO_3)_2$ or $Ni(NO_3)_2$, the PtCo or PtNi alloys could also be obtained. Similar to the PtCu alloy, both the PtCo alloys (Figures 6b and S8c,d) and PtNi alloys (Figures 6c and S8e,f) showed spherical morphology, and two elements were evenly distributed throughout the PtCo or PtNi NPs.

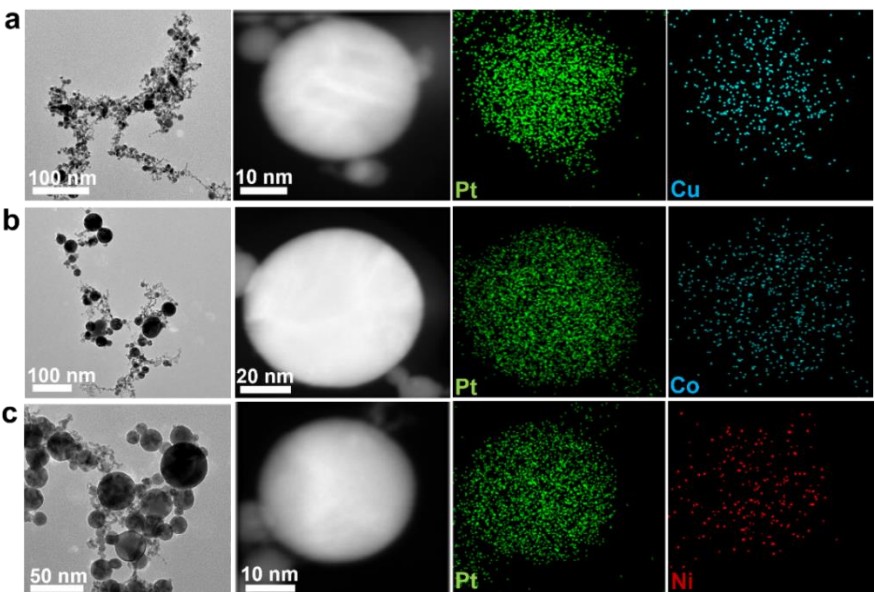

**Figure 6.** TEM and EDS mapping images of the as-prepared (**a**) PtCu, (**b**) PtCo, and (**c**) PtNi alloys.

### 2.3. Electrochemical HER Performance Measurement

The HER performance of the Pt-based bimetallic nanoalloys was measured in an Ar-saturated, 0.5 M $H_2SO_4$ solution. As shown in Figure 7a, the bulk Pt exhibited a poor HER performance with an overpotential of 72 mV to obtain the current density of 10 mA cm$^{-2}$. After alloying with Cu, Co, and Ni elements, they all exhibited a remarkably enhanced HER performance. Importantly, the PtNi alloys showed a low overpotential of 44 mV to obtain the current density of 10 mA cm$^{-2}$. The enhanced catalytic activity was also evidenced by the Tafel slope derived from the LSV curve, which is an important indicator for obtaining the reaction kinetics. In Figure 7b, the PtNi alloys displayed a low Tafel slope of 26.6 mV dec$^{-1}$, which was much lower than that of the bulk Pt (140.1 mV dec$^{-1}$), PtCo alloys (36.3 mV dec$^{-1}$), PtCu alloys (37.1 mV dec$^{-1}$), and PtAg alloys (96.3 mV dec$^{-1}$), indicating that the fast reaction was Volmer-Tafel mechanism. The lower Tafel slope means that the current density increases more rapidly as the catalyst potential decreases. In addition, the electrochemical double-layer capacitance ($C_{dl}$) was also used to investigate the electrochemically active surface area (ECSA) of the catalysts, since ECSA is proportional to $C_{dl}$. As shown in Figures 7b and S9a,b, the $C_{dl}$ of the PtNi alloy was 24.1 mF cm$^{-2}$, which was larger than that of the bulk Pt (23.1 mF cm$^{-2}$), indicating that it possessed more active sites for the catalytic reaction. Furthermore, electrochemical impedance spectroscopy (EIS) measurement was shown in Figure 7c, the PtNi alloys exhibited a smaller charge transfer resistance, suggesting the superior electrode kinetics of the PtNi alloys. Finally, long-time durability tests were employed to estimate the stability of the catalyst. As shown in Figure 7d, the current density of the PtNi alloy barely decrease after 20 h, indicating its good stability.

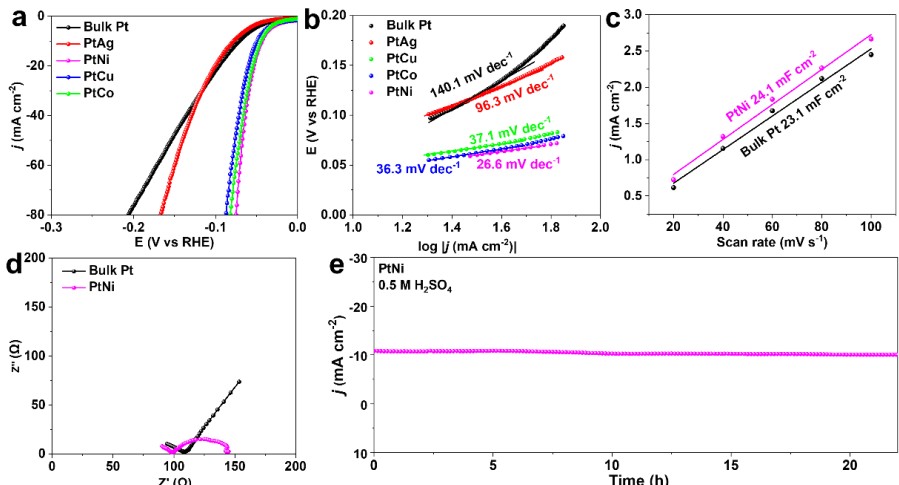

**Figure 7.** (**a**) LSV curves of the bulk Pt and PtX (X = Ag, Cu, Co, Ni) alloy, (**b**) Tafel slops curves of the bulk Pt and PtX (X = Ag, Cu, Co, Ni) alloy, (**c,d**) The $C_{dl}$ and EIS spectra of the bulk Pt and PtNi alloy, (**e**) The *i-t* curves of the PtNi alloy.

## 3. Experimental Section

### 3.1. Chemicals and Materials

All reagents were of analytical grade and further purification was not needed. $Na_2PtCl_4$, $NaHB_4$, $AgNO_3$, $Co(NO_3)_2$, $Ni(NO_3)_2$, and $Cu(NO_3)_2$ powders were purchased from Sinopharm Chemical Reagent Co., Ltd. (Shanghai, China). Ag plate with 99.99% purity was purchased from Zhongke (Hefei, China) Napu New Material Co., Ltd. (Shanghai, China). Deionized water (resistance > 18 MΩ cm$^{-1}$) was used for all experimental work.

### 3.2. Preparation of Bulk Pt

A 20 mL $Na_2PtCl_4$ (5 mg/mL) solution was added to the beaker, then, 0.715 g of $NaHB_4$ was dissolved into the 55 mL deionized water. Finally, the above solutions were mixed and heated to 343 K for 3 h. The obtained samples were washed with deionized water and then dried in an oven for subsequent use.

### 3.3. Preparation of PtX (X = Ag, Cu, Co, Ni) Alloys, and Pt Colloids

A total of 2 mg of bulk Pt was dissolved into 15 mL of deionized water and sonicated for 20 min. Then, 2 mg of $X(NO_3)_2$ (X = Ag, Cu, Co, Ni) was added to the above solution and stirred for 10 min. Finally, an Nd:YAG pulse laser with the excitation wavelength at 532 nm (pulse energy of 250 mJ, repetition of 50 Hz, and pulse duration of 7 ns) was unfocused on the above solution. After 5 min, the PtX (X = Ag, Cu, Co, Ni) alloys were obtained. The process of obtaining the Pt colloids is the same as that of the PtX (X = Ag, Cu, Co, Ni) alloys, except that no $X(NO_3)_2$ (X = Ag, Cu, Co, Ni) solution was added.

### 3.4. Preparation of PtAg Alloy NPs

A total of 2 mg of bulk Pt was dissolved into 15 mL of deionized water and sonicated for 20 min. Then, 2 mL of Ag colloids (30 ppm) were added to the above solution and stirred for 15 min. Finally, an Nd:YAG pulse laser with the excitation wavelength at 532 nm (pulse energy of 250 mJ, repetition of 50 Hz, and pulse duration of 7 ns) was unfocused on the above solution. After 5 min, the PtAg alloy NPs were obtained.

### 3.5. Preparation of Ag Colloids

Before laser ablation, the Ag plate was polished with abrasive paper and then ultrasonicated with deionized water. Then, the Ag plate was installed in the bottom of the rotating Teflon vessel, which was filled with 15 mL of deionized water. An Nd:YAG pulsed laser with the excitation wavelength at 532 nm (the repetition of 20 Hz, pulse duration of 7 ns,

and pulse energy of 75 mJ) was focused on the Ag plate for ablation. After 20 min, the bright yellow Ag colloids were obtained (concentration: 30 ppm).

*3.6. Characterizations*

The morphology and microstructures of the as-prepared samples were investigated using transmission electron microscopy (TEM) (FEI Tecnai TF20). X-ray diffraction (XRD) patterns of all samples were recorded using a Rigaku X-ray diffractometer (G2234) with Cu K$\alpha$ radiation. The X-ray photoelectron spectroscopy (XPS) was measured using a Thermo ESCALAB 250 with an Al K$\alpha$ X-ray photoelectron spectrometer at 150 W. The optical absorption spectrum of the product was measured using an ultraviolet-visible (UV-Vis) spectrophotometer (Shimadzu UV-2550).

*3.7. Electrochemical Measurements*

Electrochemical tests were performed on the CHI 660E electrochemical workstation (Shanghai Chenhua Instrument Corporation, Shanghai, China) in a conventional three-electrode system, with Ag/AgCl electrode as the reference electrode and a carbon rod as the counter electrode. All potentials mentioned below are based on a reversible hydrogen electrode (RHE). The formula for potential conversion was E (V vs. RHE) = E (V vs. Ag/AgCl) + 0.197 + 0.059 $\times$ pH. The working electrodes for HER were made by applying catalyst ink onto the glassy carbon electrode (GCE) with an area of 0.07065 cm$^{-2}$. The catalyst inks were prepared by mixing 1.8 mg powder and 6 mg carbon black in a solution containing 1 mL of isopropanol and 10 μL of 5 wt% Nafion solution using sonication for 0.5 h. Then, 10 μL of ink was dropped on the GCE for drying in the air. The HER tests were carried out in an Ar-saturated 0.5 M $H_2SO_4$ solution. Linear sweep voltammetry (LSV) was used to test the HER activity of the catalyst with a scan rate of 5 mV s$^{-1}$. CV measurements were performed for over 100 cycles (scan rate: 20 mV s$^{-1}$) to reach a stable state. Electrochemical impedance spectroscopy (EIS) was carried out at $-0.044$ V vs. RHE with a frequency from 0.01 to 100 KHz. All LSV curves were corrected with iR-compensation, and the compensation level is 90%.

**4. Conclusions**

In summary, we have demonstrated a unique laser-induced, photochemical reaction for synthesizing PtX (X = Ag, Cu, Co, Ni) alloy NMs in liquids. The relative experimental tests show the successful formation of Pt-based bimetallic alloys starting from a mixture of bulk Pt and the relevant precursors. Importantly, to investigate their potential use, these alloys were used as electrocatalysts for the HER. Especially, the PtNi alloy exhibited higher HER activity (44 mV at 10 mA/cm$^{-2}$) in the acidic media, which is much higher than that of the untreated bulk Pt (72 mV). In general, our work provides a detailed description of the bimetallic alloy alloying process and unique insights into the synthesis of bimetallic alloys, which provides guidance for the future development of bimetallic alloy electrocatalysts.

**Supplementary Materials:** The following supporting information can be downloaded at: https://www.mdpi.com/article/10.3390/catal13061018/s1, Figure S1: (a,b) XPS spectrum of Pt and PtAg alloys, (c) C1s of Pt and PtAg alloys, (d) the auger electron spectrum of Ag for PtAg alloys; Figure S2: The Ag NPs were obtained by irradiating the AgNO$_3$ solution with laser irradiation; Figure S3: UV-vis absorption spectra of AgNO$_3$ solution irradiated by laser; Figure S4: UV-vis absorption spectra of AgNO$_3$ solution irradiated by a 300W Xenon lamp with a wavelength of 520 nm; Figure S5: The pH of AgNO$_3$ solution (125 mg/L) (a) initial (b) after 5 min of laser irradiation. The pH of the intermixture of bulk Pt and AgNO$_3$ solution (125 mg/L) (c) initial (d) after 5 min of laser irradiation; Figure S6: TEM images and histogram of size distribution of Ag NPs; Figure S7: (a) TEM image of PtAg alloy NPs, (b) the HRTEM image of PtAg alloy NPs, (c,d) EDX line scans and EDS mapping of PtAg alloy NPs; Figure S8: EDX line scans and EDX energy spectrum of PtX (X = Cu, Co, Ni) alloy NMs, (a,b) PtCu alloys, (c,d) PtCo alloys, (e,f) PtNi alloys; Figure S9: Cyclic voltammetry curves at scan rates ranging from 20 to 100 mV s$^{-1}$ for (a) the bulk Pt, (b) PtNi alloy.

**Author Contributions:** Conceptualization, T.H. and C.L.; methodology, T.H., Y.F. and Y.C.; validation, T.H., Y.M. and Y.Y.; formal analysis, Y.F. and P.L.; investigation, P.L.; resources, C.L. and P.L.; data curation, Y.Y., Y.F., T.H. and J.L.; writing—original draft preparation, T.H.; writing—review and editing, Y.F. and T.H.; visualization, T.H.; supervision, T.H.; project administration, C.L.; funding acquisition, C.L. All authors have read and agreed to the published version of the manuscript.

**Funding:** This work was financially supported by the National Natural Science Foundation of China (NSFC, 51971211, 52071313), Youth Innovation Promotion Association of CAS (2017483), the Collaborative Innovation Program of Hefei Science Center of CAS (Grant 2021HSC-CIP015), Plan for Anhui Major Provincial Science & Technology Project (No. 202103a05020015), and the HFIPS Director's Fund, Chinese Academy of Sciences (Nos. YZJJ202102).The authors would like to thank Shiyanjia Lab (www.shiyanjia.com) for the XPS measurement.

**Institutional Review Board Statement:** Not applicable.

**Informed Consent Statement:** Not applicable.

**Data Availability Statement:** The data presented in this study can be obtained from the first author.

**Conflicts of Interest:** The authors declare no conflict of interest.

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
