# Peer review of "Laser Irradiation-Induced Pt-Based Bimetallic Alloy Nanostructures without Chemical Reducing Agents for Hydrogen Evolution Reaction"

_catalysts, doi:10.3390/catal13061018_

Round 1

Reviewer 1 Report

Review Report

The manuscript “Laser irradiation-induced alloying in liquids for Pt-based bimetallic Alloy Nanostructures in the absence of chemical reducing agents” reports a straightforward approach to preparing bimetallic nanoparticles in liquids by employing laser without additional chemicals. Sufficient data have been provided in this manuscript to evidence the effectiveness of this method to prepare diverse bimetallic nanoparticles. In addition, the growth mechanism and the HER performances of these Pt-based bimetallic nanoparticles have been discussed. This work would shed light on applying laser energy to assist the synthesis of metallic NPs with well-controlled structures and properties desirable for applications, e.g., photo electrocatalysis, which would be of interest to the readership of Catalysts. However, the current manuscript seems to be poorly organized and written, especially, since the description and discussion of the data are difficult to understand. Format errors are also common in the manuscript. Below are the concerns about the manuscript.

1.       Figure 1, Line 158, what is the phase of the twin boundary? Or what do you want to prove by the existence of the twin boundary?

2.       Figure 2a, compared with Pt (blue line Figure 2a), the metallic Pt0 peaks have a negative shift in PtAg alloy, rather than a positive?

3.       Figure 3, the colors indexed Ag and Pt are suggested to be consistent.

4.       Figure 4, what do you want to say by presenting the UV-Vis absorption spectra of Pt colloid?

5.       About the formation mechanism of PtAg alloy, do you have data to show that the bulk Pt has been broken into smaller parts? How do you know that the vapor pressure is about 108 Pa?

6.       Figure 7a should be checked with the wrong axes.

7.       The word “monumental” in Line 38 seems wrong. Should it be “mono-elemental”?

8.       In Sections 2.3 and 2.4, Lines 98, and 106, is it “unfocused ” or “focused” on…?

9.       The numbers of Sections (Characterization and Electrochemical measurements) are wrong?

10.   Line 116-121, the description sounds very weird.

11.   Line 125, “Shanghai”, rather than “Shang hai”.

12.   Line 137, “the potential of 10 mA cm-2” is wrong.

Reviewer 2 Report

 The manuscript "Laser Irradiation-Induced Alloying in Liquids for Pt-Based Bi-2 Metallic Alloy Nanostructures in the Absence of Chemical Reducing Agents" presents an approach for producing bimetallic alloys and particles of noble and transition metals for catalyst applications.

The study is definitely of interesting and will be useful for scientists in the field of laser materials science and catalysis. However, some issues have to be revised before the manuscript can be accepted.

- Firstly, the authors are invited to reconsider the title of the manuscript, as in its current form it sounds very long and does not fully describe the research carried out. Also, the terms "absence" should be reviewed.

- It is highly recommended to avoid using high-sounding statements like "new", "first time", "new strategy", "new direction", etc. in the Abstract and in the Conclusion. Also, the first sentence in the Abstract seems more suitable for the Introduction, and the authors should be careful about this.

- In the section Introduction, it is useful to focus on other approaches using different lasers and other precursors in order to make an relevant comparison. For example, these articles or any others could be considered if the authors prefer. (https://www.mdpi.com/1996-1944/13/15/3359 ; https://www.mdpi.com/2079-4991/12/1/146 )

- The formation of silver nanoparticles by light is a well-studied field. The mechanism of water decomposition described by the authors has been previously reported in numerous papers (e.g. https://onlinelibrary.wiley.com/doi/10.1002/adom.202201094) and it is also worth mentioning. Further, the fact that the decomposition of Silver Nitrate follows a photochemical mechanism and that the decomposition of the other salts probably follows a thermal scenario is also interesting and the authors are invited to speculate in this direction or to provide additional measurement.

- the authors are successful in providing additional studies on the decomposition of silver nitrate, but in order to enhance the manuscript it can be recommended to measure the temperatures in the focal zone as well as the energy of the laser radiation during the experiments.

 - 284 - overpotrntial (typo)

- The sentence "Simultaneously, the bulk Pt is broken into Pt atoms which keep zero chemical state and alloy with Ag atoms" does not look correct because there is no direct proof of the decomposition exactly into Pt atoms.

- Figure 3 and the explanation offered by the authors looks a bit far-fetched as the individual areas are considered and the distribution can be quite random regardless of the time of exposure. Thus, the absorption spectra in Fig. 4 precisely demonstrates that the dependencies from 1 to 5 minutes are almost negligible. 

- The results of the pH measurement shown in the SI should be supplemented by data on the initial concentration of metal salts, as there is doubt that such low values of the pH cannot be achieved in aqueous solutions.

Round 2

Reviewer 1 Report

The authors have addressed the concerns raised in the first review. I would support the publication of this revised manuscript.